# DNA Barcoding and *ITS2* Secondary Structure Predictions in Taro (*Colocasia esculenta* L. Schott) from the North Eastern Hill Region of India

**DOI:** 10.3390/genes13122294

**Published:** 2022-12-05

**Authors:** Mayengbam Premi Devi, Madhumita Dasgupta, Sansuta Mohanty, Susheel Kumar Sharma, Vivek Hegde, Subhra Saikat Roy, Rennya Renadevan, Kinathi Bipin Kumar, Hitendra Kumar Patel, Manas Ranjan Sahoo

**Affiliations:** 1Indian Council of Agricultural Research (ICAR) Research Complex for North Eastern Hill Region, Imphal 795004, India; 2College of Agriculture, Central Agricultural University (CAU-Imphal), Kyrdemkulai 793105, India; 3Central Horticultural Experiment Station, ICAR–Indian Institute of Horticultural Research, Bhubaneswar 751019, India; 4ICAR—Indian Agricultural Research Institute, Pusa Campus, New Delhi 110012, India; 5ICAR—Central Tuber Crops Research Institute, Thiruvananthapuram 695017, India; 6ICAR—Indian Institute of Horticultural Research, Bengaluru 560089, India; 7Centre for Cellular and Molecular Biology, Hyderabad 570007, India

**Keywords:** *Araceae*, DNA barcoding, genetic discrimination, *ITS2* secondary structure, molecular phylogeny, taro

## Abstract

Taro (*Colocasia esculenta* L. Schott, *Araceae*), an ancient root and tuber crop, is highly polygenic, polyphyletic, and polygeographic in nature, which leads to its rapid genetic erosion. To prevent the perceived loss of taro diversity, species discrimination and genetic conservation of promising taro genotypes need special attention. Reports on genetic discrimination of taro at its center of origin are still untapped. We performed DNA barcoding of twenty promising genotypes of taro indigenous to the northeastern hill region of India, deploying two chloroplast-plastid genes, *matK* and *rbcL*, and the ribosomal nuclear gene *ITS2*. The secondary structure of *ITS2* was determined and molecular phylogeny was performed to assess genetic discrimination among the taro genotypes. The *matK* and *rbcL* genes were highly efficient (>90%) in amplification and sequencing. However, the *ITS2* barcode region achieved significant discrimination among the tested taro genotypes. All the taro genotypes displayed most similar sequences at the conserved *matK* and *rbcL* loci. However, distinct sequence lengths were observed in the *ITS2* barcode region, revealing accurate discriminations among the genotypes. Multiple barcode markers are unrelated to one another and change independently, providing different estimations of heritable traits and genetic lineages; thus, they are advantageous over a single locus in genetic discrimination studies. A dynamic programming algorithm that used base-pairing interactions within a single nucleic acid polymer or between two polymers transformed the secondary structures into the symbol code data to predict seven different minimum free energy secondary structures. Our analysis strengthens the potential of the *ITS2* gene as a potent DNA barcode candidate in the prediction of a valuable secondary structure that would help in genetic discrimination between the genotypes while augmenting future breeding strategies in taro.

## 1. Introduction

Taro (*Colocasia esculenta* L. Schott), which belongs to the *Araceae* family, is one of the ancient tuber crops and the most widespread food crop cultivated in the tropical to temperate world [1]. In ancient civilizations, the paddy crop was considered as weed in a flooded taro field. Today, paddy has become the staple food crop in the developed world. However, taro is still considered an orphan crop and a crop for the resource-poor [2]. Taro corms are one of the cheapest sources of carbohydrates, protein, minerals, and vitamins and are used in treatment of fungal infections, pulmonary congestion, tuberculosis, and ulcers [3]. Taro is believed to have originated in Indian subcontinent, and the north eastern hill (NEH) region of India, Indo-Myanmar in particular, is considered as the secondary place of origin of taro [4]. It was further dispersed to East and Southeast Asia, the Pacific Islands, Africa, and tropical America [5]. Taro is highly polymorphic, polygenetic, and polygeographic in nature. The phylogeographic and phylogenetic evolution of taro is still unclear [6], which makes this potential tuber crop underutilized.

The wild taro populations were widespread worldwide before its cultivation started, thus showing huge genetic diversities across geographical regions [7]. The NEH region of India is a treasure trove for taro diversities with immense genetic potential for biotic and abiotic stress tolerance. There is a scope for introgression of this untapped gene pool for the genetic improvement of taro. The rich taro diversities in the NEH region of India need special attention for the conservation of the gene pool and the prevention of perceived genetic erosion [8]. Proper identification, characterization of promising landraces, and understanding of their genetic background facilitate rapid genetic improvement in taro [9]. Several marker-assisted studies have elucidated the genetic diversity in taro; however, it is difficult to distinguish the progenitor and descendants of taro’s wild relatives due to their polygenic nature [10].

Phenotypic indicators have resolved the morphological variability with baseline information on diversity. However, morphological indicators could not confer genetic variations due to spontaneous vegetative mutagenesis [11]. Molecular characterization using simple sequence repeat (SSRs) has been used to study the genetic diversity of taro worldwide [6] and understand its genetic bottleneck and secondary domestication. Genetic characterization of the taro gene pool at the chloroplast DNA (cpDNA) using restricted fragment length polymorphism (RFLP) was attempted in wild and cultivated genotypes, which formed a monophyletic group without distinct clade structures [12]. Previous attempts could not establish a phylogenetic relationship between the wild and cultivated taro gene pool [1].

Recent advancements in molecular and *omics* tools have helped overcome the limitations of morphological discriminations [13]. DNA barcoding is an advanced, robust, and efficient tool that enables species identification and delimitation [14]. DNA barcoding involves two important steps, including building the barcode library from known taxa and assigning the barcode sequence against the library for species identification following phylogenetic relationships [15]. DNA barcoding and molecular phylogeny efficiently discriminate among closely related species [16]. On the other hand, RNA secondary structures predicted from the nuclear *ITS2* sequences could validate the phylogenetic relationships derived from the short DNA sequences. RNA molecules are the basic building blocks that fold into secondary and tertiary structures with diverse functional properties. RNA structural motifs can unveil the genetic understanding of how RNA structures regulate functional elements [17]. RNA secondary structure predictions can help in understanding the RNA molecules that govern gene function and regulation. RNA secondary structure based on the minimum free energy (MFE) algorithm has higher accuracy in establishing relationships among the RNA families [18], eliminating pseudogene footprints and minimizing sequencing error in DNA barcoding [19].

A DNA barcode deploys a short gene sequence from a conserved coding and non-coding region used as a marker for species discrimination [20]. Chloroplast genome regions are highly conserved regions due to the complex uniparental inherent repetition mechanisms and are often used for phylogenetic studies [21]. The chloroplast-plastid gene *matK* encodes the maturase K enzyme [22], whereas the *rbcL* gene encodes the ribose-1,5-bisphosphate carboxylase oxidase enzyme [23]. Although there are some mixed responses in terms of amplification and sequence success rate, the *matK* gene is one of the most informative loci for establishing phylogenetic relationships [24]. The *rbcL* gene is the first single-copy gene sequenced in plants contains no intron [25]. As one of the most conserved genes in the chloroplast region, *rbcL* is widely used to establish phylogenetic relationships among the species showing too little variation at the species level [25]. However, *ITS2,* the internal transcribed spacer region of the nuclear-ribosomal region is often considered as a candidate barcode gene due to its efficient amplification ability, universal characteristics, and significant variability [26]. Combining barcode loci efficiently elucidates intra-specific variability. CBOL (2009) [20] suggests that a combination of multi-loci barcode regions (*matK*+*ITS*) apparently discriminates the species more efficiently than a single-locus region (*matK*+*rbcL*). However, the distinct variation in the *ITS* region supersedes the ambiguity that arises among the variation in the chloroplast region [27]. Characterization studies of the intra-specific variations in taro advanced breeding lines indigenous to its centre of origin, the northeastern hill region of India, using multiple barcode regions are particularly scarce [3,28,29].

The present study aimed to investigate genetic discrimination using the conserved *matK*, *rbcL*, and *ITS2* barcode genes among the twenty advanced breeding gene pools of taro developed from the landraces indigenous to the NEH region of India. In addition, we assessed the efficacy of the nuclear-plastid and nuclear region barcode markers for species delimitation among the taro genotypes. The molecular phylogeny based on the DNA barcode sequences confirms the inter-relationship among the advanced breeding lines at the conserved region. The present study also focused on MFE-based RNA secondary structure predictions concerning the *ITS2* nuclear gene for further validation of the molecular phylogeny by grouping the genotypes into various clades. The DNA barcode-anchored species delimitation would be useful for a close understanding of the genotypes, helping to augment future breeding strategies in taro.

## 2. Materials and Methods

### 2.1. Plant Materials and Experimental Site

Twenty diverse taro genotypes (Appendix A) were collected from the northeastern hill (NEH) regions of India and from the Indian Council of Agricultural Research (ICAR)-Central Tuber Crops Research Institute (CTCRI), Regional Centre, Bhubaneswar, India. The genotypes were maintained at the Langol hill research farm of the ICAR-Research Complex for the Northeastern Hill Region (ICAR RC NEHR), Manipur Centre, Imphal, India. The experimental site is located at 24° 50′ N latitude, 93° 55′ E longitude, and at an altitude of 860 m above mean sea level. The taro genotypes were grown following the package of practices recommended by the ICAR-CTCRI, Regional Centre, Bhubaneswar, India. The climatic condition of the study site is tropical with an average temperature of 22–30 °C, rainfall of 1540 mm, and a relative humidity of 85–90% during the crop growth period from April to October 2019. This study was conducted under the twinning program of the Department of Biotechnology (DBT), Ministry of Science and Technology, Govt. of India, in collaboration with the ICAR-CTCRI, Thiruvananthapuram, India, with prior approval from the competent authorities which ensure compliance with ethical regulations.

### 2.2. Plant Sample Preparation and Total Genomic DNA Isolation

The central newly emerged leaf samples from the twenty taro genotypes were collected in sterile Eppendorf tubes and snap-frozen in liquid nitrogen for genomic DNA (gDNA) extraction. The collected leaf tissue was homogenized by using Tissue Lyser LT (QIAGEN, Hilden, Germany) and processed for total gDNA isolation in a QIAcube automated nucleic acid extractor (QIAGEN, Hilden, Germany) using a DNeasy Plant Mini Kit (QIAGEN, New Delhi, India) and following the manufacturer’s protocol. The quantification of the isolated gDNA was carried out by using a QIA expert, single-channel UV–Vis nano reader (QIAGEN, Hilden, Germany) and electrophoresed on a 0.8% agarose gel (Tarson, Mumbai, India). To enable PCR, the final concentration of the gDNA was adjusted to 50 ng µL^−1^ by dilution from a total gDNA concentration of 40-120 ng µL^−1^.

### 2.3. PCR Amplification and Purification

PCR amplification was carried out with the *matK*, *rbcL*, and *ITS* DNA barcode primers as recommended by the Consortium for the Barcode of Life–Plant Group [20]. The *matK*, *rbcL,* and *ITS2* primers were synthesized by M/S Bioserve Biotechnologies (Hyderabad, India) Pvt. Ltd., Hyderabad, India. The primer details are given in Table 1. Each primer set was optimized by gradient PCR to obtain the annealing temperature. The PCR reaction was carried out in a volume of 25 µL consisting of 50 ng template DNA (1 µL), 12.5 µL 2 × Taq PCR master mix (QIAGEN, New Delhi, India), and 10 pM primers. The amplification was achieved under the following conditions: an initial denaturation at 95 °C for 5 min, followed by 40 cycles of denaturation at 95 °C for 1 min, annealing at 55 °C for 1 min, extension at 72 °C for 1 min, and a final extension at 72 °C for 10 min in a SimpliAmp^TM^ thermal cycler (Applied Biosystems Life Technologies^®^, Waltham, MA, USA). For the detection of amplicons, PCR products were resolved on a 1.5% agarose gel, and pictures were taken on the E-box gel documentation imaging system (Vilber Lourmat, France). The PCR products were purified using a QIA quick PCR purification kit (QIAGEN, New Delhi, India) by following the manufacturer’s instructions. The quality of the purified PCR products was further processed for DNA sequencing at M/S Bioserve Biotechnologies (India) Pvt. Ltd., Hyderabad, India.

### 2.4. DNA Sequencing

The sequencing of the purified PCR products was performed at M/S Bioserve Biotechnologies (India) Pvt. Ltd., Hyderabad, India, using Sanger sequencing (ABI Genetic Analyzer 3730, 48 capillaries, 50 cm, Thermo Fisher, Waltham, MA, USA). The obtained sequences were viewed and analyzed in FinchTV v1.4.0 (Geospiza, Denver, CO, USA).

### 2.5. Bioinformatics Analysis

The reads from forward and reverse sequences from all the PCR products amplified with *matK*, *rbcL*, and *ITS* primers were trimmed using SnapGene v5.3 (https://www.snapgene.com; accessed on 1 October 2022). The manual editing of barcode gaps was done in pairwise alignment view using BLAST, and by following nucleotide blast with the maximum similarity score and lowest E value, species identification was performed. The muscle algorithm method was carried out using ClustalW v10.1.8 (https://www.genome.jp/tools-bin/clustalw; accessed on 1 October 2022) to obtain multiple sequence alignment (MSA) using MEGA11 software (Molecular Evolutionary Genetic Analysis; https://www.megasoftware.net; accessed on 1 October 2022) [30]. By considering the transitional and transversional nucleotide substitution, the phylogenetic study was executed using the neighbor-joining tree (NJT) and minimum evolution method (MEM) with 1000 as the “*Bootstrap phylogeny*” value and the “*kimura–2–parameter”* substitution model (*d–transitions*) in MEGA11 software [31].

### 2.6. RNA Secondary Structure Predictions

The nucleotide sequences obtained from *ITS2* primers were used to predict the *RNA* secondary structure from the rRNA database of RNAfold Web Server v2.4.18 (http://rna.tbi.univie.ac.at/cgi-bin/RNAWebSuite/RNAfold.cgi; accessed on 1 October 2022) [32].

## 3. Results

### 3.1. PCR Amplification, Sequencing, and Multiple Sequence Alignment

The PCR amplification rate was higher (>90%) for the *matK* and *rbcL* genes as compared to the *ITS2* genes (>80%) [Table 1]. The *matK*, *rbcL*, and *ITS2* sequences of the twenty taro genotypes were queried against the NCBI database (https://www.ncbi.nlm.nih.gov; accessed on 1 October 2022) to find the regions of similarity, and the blast-edited sequences were processed further for analysis. The consensus lengths of *matK* (222–514 bp), *rbcL* (250–270 bp), and *ITS2* (313–315 bp) sequences after trimming varied among the taro genotypes. The distribution of mean nucleotide base frequencies of candidate nucleotide sequences at different coding positions (*matK*, *rbcL*, and *ITS2*) in taro genotypes are presented in Table 2. Sequence analysis of the *matK*, *rbcL*, and *ITS2* loci identified the twenty tested taro genotypes as *Colocasia esculenta* with 99.04 (*ITS2)*–100% (*matK* and *rbcL*) homology among all the taro genotypes (Table 3). The *ITS2* gene exhibited wide dispersal sequence similarity, as revealed by the sequence alignments, whereas *matK* and *rbcL* exhibited homologous sequences among the tested taro genotypes.

The rate of different transitional substitutions and transversional substitutions showed significant variations among the different conserved barcode regions, which indicated the differences among the taro genotypes (Table 4). *ITS2* showed higher transitional substitution (point mutations) values among the nucleotides across the regions compared to *rbcL* and *matK*. However, *matK* exhibited higher transversional substitution values followed by *rbcL* and *ITS2* (Table 4).

### 3.2. Phylogenetic Studies, Replacement Rate Matrices, and DNA Barcoding

The unrooted maximum likelihood tree (MLT) [Figure 1, Figure 2 and Figure 3] depicts the phylogenetic relationship with the branch lengths representing evolutionary distances or the *bootstrap* values in a *K2P* model. The unrooted MLT derived from *matK* sequences grouped the twenty taro genotypes into two major clades. Clade I comprises 14 genotypes, and clade II includes 6 genotypes (Figure 1). The *rbcL* sequences showed 100% homogeny among the tested genotypes, and as such could not establish significant discrimination in the present study (Figure 2). However, the ribosomal-nucleus *ITS2* gene sequences categorized the taro genotypes into five distinct clades. Clades I, II, and III each comprise five genotypes, whereas clades IV and V include two and three genotypes, respectively (Figure 3).

### 3.3. ITS2 Secondary Structure Predictions

A dynamic programming algorithm transformed the secondary structures into the symbol code data to predict the minimum free energy secondary structure. The *ITS2* loci displayed seven different e-loop motif features, which contained a ring-pin that was a typical feature of the secondary structure model of taro (Figure 4). Structures I, III, V, and VI represented a centroid ring with 3-6 hair-pin motifs and 2-6 bulge loops. However, structures II, IV, and VII displayed a centroid bulge with multi-branched loops (Figure 4). As per the *ITS2* secondary structures, the taro genotypes were grouped into seven different clades. Clade I comprises six genotypes (RCMC-1, RCMC-2, DP-25, TSL, BBSR, Satasankha), Clade II includes five genotypes (RCMC-5, Duradim, Jhankri, Topi 1kR, BBSL), and Clade III involves five genotypes (RCMC-6, P. Chandel, R5MK, R1B9, PCT1). However, the RCMC-10, Topi, MK, and R5JH10 genotypes exhibited unique secondary structures each. R5JH10 exhibited a similar structure to that of Clade I with the omission of the helix at the centroid ring (Figure 4). The *ITS2* secondary structures are the unique genetic structures at the conserved nuclear region which could be useful in discriminating the taro landraces at the genotypic level by understanding the RNA molecules.

## 4. Discussion

Tropical root crops and the products they generate are crucial components for a sustainable agricultural system as well as key components of human and animal nutrition. The domestication of numerous perennial herbaceous species in the *Colacasia* genus calls for the use of various evolutionary biology approaches that can effectively identify and authenticate the species. Additionally, a comparison of the physiological, biochemical, and chromosomal (heritable) traits of several taro species found discrepancies in the categorization made within the genus, indicating the need for more trustworthy methods involving different genetic DNA markers with broad binomial nomenclature specificity.

### 4.1. DNA Barcoding Using rbcL, matK, and ITS2 Genes

The adoption of multi-loci techniques [33] including both nuclear and chloroplast DNA barcodes has greatly improved accuracy of species identification, especially for closely related species, despite the absence of one universal barcode that can be utilized to identify relevant plant species [21]. The molecular identification of the *Colacasia* species was made possible by the nucleotide sequences of plastid genes, the inter-gene region (*rbcL,* and *matK*), and the nuclear region (*ITS2*). However, the *ITS2* sequences have a distinct way of differentiating the taro cultivars and do so more efficiently.

We used DNA barcoding analysis employing the extremely diverse *ITS2* barcoding regions to confirm the accuracy of species identification for the indicated taro specimens placed in regional NCBI banks. Even though there is extensive species coverage of all the DNA barcodes in the NCBI gene bank archive for many species of plants, such as *Colocasia esculenta*, *ITS2* enables more precise species categorization using sequence similarity (>99%). In order to better comprehend the features of the taxonomic variations of taro farmed in Brazil, a phylogenetic study was carried out, and the variability of taro was investigated by [29] using chloroplast genome sequences, such as *rbcL* and *pbsA-trnH*. In [34], the genetic diversity of taro was assessed using two chloroplast DNA barcoding markers, *rbcL* and *pbsA-trnH,* and the authors found an enhanced perception of genetic evaluation and of their characteristics. DNA barcoding analysis employing the extremely variable chloroplast *trnL* and nuclear *ITS2* DNA barcoding regions has been used to confirm the correctness of species identification for indicated Eurasian *Vicia* species [35]. However, [36] acknowledge that after a comparison of several variations in the number of species detected from phylogenetic groupings,, the genetic diversity index using the *rbcL* marker was unable to resolve the identification of yam (*Dioscorea spp.*) well, exposing the ineffectiveness of this marker in differentiating the genetic variability of yam.

Using the results of the BLAST analysis of the DNA sequences, the twenty studied genotypes were grouped into the same species with notably high correlation values, and have been identified with >99% identity (Table 1 and Table 2). Furthermore, we have demonstrated that diversity studies in the taro can rely on the nuclear *ITS2* region. Although minor differences were found for a few clades that resulted from the *rbcL* and *matK* phylogenetic trees, we confirmed the topology of the genotypes and the related reference records were found to be consistent (Figure 1 and Figure 2). Although *rbcL* and *matK* are not the best identifiers for species discrimination, the relatively conserved nuclear *ITS2* region was considered the finest marker because it allows for a more ideal interspecific alignment.

### 4.2. ITS2 Marker and RNA Secondary Structure

In order to provide insight into emerging RNA control frameworks, digitally generated RNA structures should reflect the native RNA folded state of a species. Because it limits sequencing inaccuracy and removes pseudogene imprints, RNA secondary initial design is an effective approach for species characterization [37], RNA structure is constantly found to be more conserved than the DNA sequences obtained. To the standard approach of sequence-based alignment, the homologous RNA secondary structures can operate to function equally on the binding of the base pair, and cleaving or catalyzing may be considered to yield unique nucleotides. Hence, base-pair interactions resulting in secondary or tertiary RNA structures are necessary for *ITS2* to function [38]. The accuracy of the results in identifying species under common clades or groups can be considerably increased if the common structures of these homologous sequences are measured during RNA folding and alignment. On the other hand, determining a set of RNA molecules’ similar structures can be used to anticipate the essential components of their function [39].

In particular, the *ITS2* preserved structure has recently received a lot of attention and application [18]. *ITS2*-derived RNA folding and sequence-structure alignment have increasingly utilized *ITS2* sequences and their established secondary structures as templates. Our discovered RNA secondary structure was compared to all the genotypes, which revealed a “ring-pin” model structure as the general core throughout all taro genotypes, providing a superior differentiation method than other markers, such as *rbcL* and *matK*. Therefore, the *ITS2* marker is found to be an excellent alternative identifying method for integrated analysis of RNA structures and their function in evolutionary attention. Notably, the *ITS2* DNA barcoding marker will be taken forward for frequent use in phylogenetic techniques that will grow exponentially and it will become a sequence-directed genetic information contributor. The identification of these RNA secondary patterns increases the utility of the *ITS2* secondary structure in taro as a taxonomic DNA marker. Breeders of taro may select from a variety of parents for greater diversity or a hybrid for taro crop development [3]. Since the *ITS2* marker is revealed to be effective and it can be recommended as the stand-alone marker of choice for DNA barcoding of *Colocasia* accessions, especially for accurate identification, discrimination, and estimation of the genetic diversity of this important tuber crop.

## 5. Conclusions

This study described the efficiency of DNA barcode genes in chloroplast-plastid (*matK* and *rbcL*) and ribosomal-nuclear (*ITS2*) regions in discriminating twenty promising taro lines at the genotypic level. Although *matK* and *rbcL* markers exhibited higher amplification and sequencing rates, *ITS2* emerged as an ideal marker for DNA barcoding, phylogenetic relationship determination, and secondary structure predictions in our study. The results of this study successfully delimit the *ITS2* boundary among the taro genotypes and the associated evolutionary information. The divergent *ITS2* sequence alignments have been used for establishing the phylogenetic relationship and secondary structure predictions. The *ITS2* loci-based maximum likelihood tree grouped the twenty taro genotypes into five major groups; however, *ITS2* secondary structure predictions categorized the twenty genotypes into seven different clades. The close relationship of the taro genotypes indicated that they homogenously originated from the same geographical region. This study is the first report on genetic discrimination of taro genotypes from the center of origin of taro, the northeastern hill region of India, in particular. This study also provides insight into the scope of the development of species-specific markers for the discrimination of closely related taxa by deploying multiple barcode genes. Understanding the RNA molecules would be useful for the selection of parents in trait-specific breeding strategies for taro improvement.

## Figures and Tables

**Figure 1 genes-13-02294-f001:**
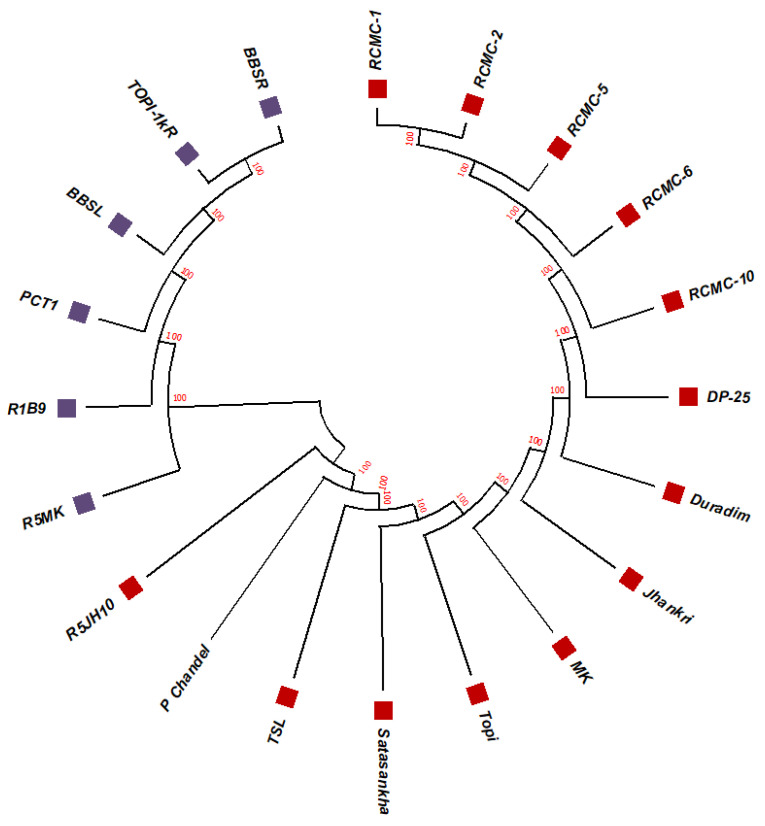
The unrooted maximum likelihood tree of the twenty taro genotypes based on the *matK* gene sequences (Note: Bootstrap scores of ≥50% only are shown for each branch after 1000 bootstrap replication tests).

**Figure 2 genes-13-02294-f002:**
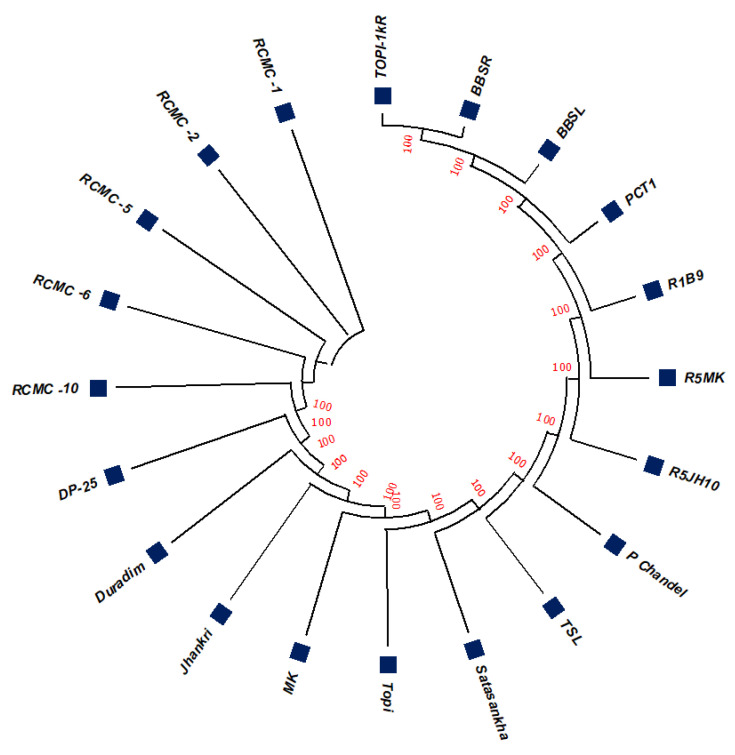
The unrooted maximum likelihood tree of the twenty taro genotypes based on the *rbcL* gene sequences (Note: Bootstrap scores of ≥50% only are shown for each branch after 1000 bootstrap replication tests).

**Figure 3 genes-13-02294-f003:**
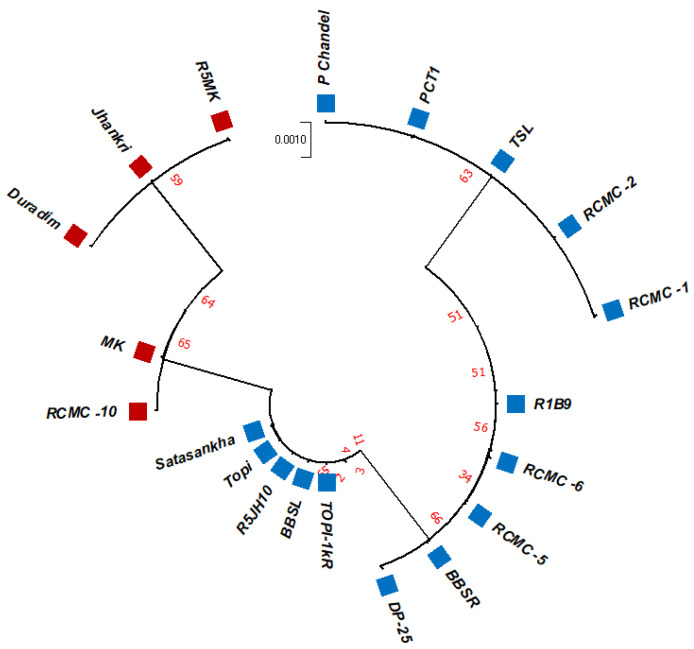
The unrooted maximum likelihood tree of the twenty taro genotypes based on the *ITS2* gene sequences (Note: Bootstrap scores of ≥50% only are shown for each branch after 1000 bootstrap replication tests).

**Figure 4 genes-13-02294-f004:**
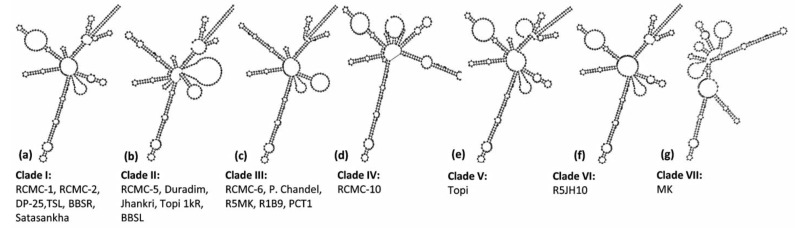
Seven consensus *ITS2* secondary structure predictions from the twenty taro genotypes based on the minimum free energy (MFE).

**Table 1 genes-13-02294-t001:** Details of the barcode primers used for PCR amplification of the taro genotypes.

Region	Primers	Sequences (5′ to 3′)	Amplicon Size	Amplification Success Rate
*matK*	*matK XF* *matK 5R*	TAATTTACGATCAATTCATTC GTTCTAGCACAAGAAAGTCG	1000 bp	90–95%
*rbcL*	*rbcLa F* *rbcLa R*	ATGTCACCACAAACAGAGACTAAAGC GTAAAATCAAGTCCACCRCG	600 bp	90%
*ITS2*	*ITS2 S2F* *ITS2 S3R*	ATGCGATA CTTGGTGTGAATTATAGAAT GACGCTTCTCCAGACTACAAT	300–400 bp	80–85%

**Table 2 genes-13-02294-t002:** The nucleotide base frequencies of candidate nucleotide sequences at different coding positions in taro plants.

Sequences	Base Contents (%)					
	A	T	G	C	AT	GC
*matK*	30.75	37.56	12.56	19.13	68.31	31.69
*rbcL*	25.80	33.41	20.28	20.51	59.20	40.80
*ITS2*	19.52	11.44	34.78	34.26	30.96	69.04

**Table 3 genes-13-02294-t003:** BLASTn search results for *matK*, *rbcL*, and *ITS2* gene sequences of twenty taro (*Colocasia esculenta*) genotypes.

Voucher Name	*matK* and *rbcL*				*ITS2*			Species
	Percent Identity	E Value	Accession No. (*matK*)	Accession No. (*rbcL*)	Percent Identity	E Value	Accession No. (*ITS2)*	
RCMC-1	100%	0.0	LT995105.1	MH270468.1	99.05%	3 × 10^−156^	MK961250.1	*C. esculenta*
RCMC-2	100%	0.0	LT995105.1	MH270468.1	99.05%	3 × 10^−156^	MK961250.1	*C. esculenta*
RCMC-5	100%	0.0	LT995105.1	MH270468.1	99.37%	7 × 10^−158^	MK961250.1	*C. esculenta*
RCMC-6	100%	0.0	LT995105.1	MH270468.1	99.37%	7 × 10^−158^	MK961250.1	*C. esculenta*
RCMC-10	100%	0.0	LT995105.1	MH270468.1	99.36%	2 × 10^−157^	MK961250.1	*C. esculenta*
DP-25	100%	0.0	LT995105.1	MH270468.1	100.00%	1 × 10^−160^	MK961250.1	*C. esculenta*
Duradim	100%	0.0	LT995105.1	MH270468.1	99.04%	1 × 10^−155^	MK961250.1	*C. esculenta*
Jhankri	100%	0.0	LT995105.1	MH270468.1	99.36%	2 × 10^−157^	MK961250.1	*C. esculenta*
MK	100%	0.0	LT995105.1	MH270468.1	99.36%	2 × 10^−157^	MK961250.1	*C. esculenta*
Topi	100%	0.0	LT995105.1	MH270468.1	99.68%	5 × 10^−159^	MK961250.1	*C. esculenta*
Satasankha	100%	0.0	LT995105.1	MH270468.1	99.68%	5 × 10^−159^	MK961250.1	*C. esculenta*
TSL	100%	0.0	LT995105.1	MH270468.1	99.05%	3 × 10^−156^	MK961250.1	*C. esculenta*
P Chandel	100%	0.0	LT995105.1	MH270468.1	99.05%	3 × 10^−156^	MK961250.1	*C. esculenta*
R5JH10	100%	0.0	LT995105.1	MH270468.1	99.68%	5 × 10^−159^	MK961250.1	*C. esculenta*
R5MK	100%	0.0	LT995105.1	MH270468.1	99.04%	1 × 10^−155^	MK961250.1	*C. esculenta*
R1B9	100%	0.0	LT995105.1	MH270468.1	99.05%	3 × 10^−156^	MK961250.1	*C. esculenta*
PCT1	100%	0.0	LT995105.1	MH270468.1	99.05%	3 × 10^−156^	MK961250.1	*C. esculenta*
BBSL	100%	0.0	LT995105.1	MH270468.1	99.68%	5 × 10^−159^	MK961250.1	*C. esculenta*
TOPI-1kR	100%	0.0	LT995105.1	MH270468.1	99.68%	5 × 10^−159^	MK961250.1	*C. esculenta*
BBSR	100%	0.0	LT995105.1	MH270468.1	100.00%	1 × 10^−160^	MK961250.1	*C. esculenta*

**Table 4 genes-13-02294-t004:** Maximum likelihood estimates of the substitution matrix of *matK*, *rbcL,* and *ITS2* sequences. Rates of different transitional substitutions are in **bold,** and those of transversional substitutions are shown in *italics*.

*matK*					
** *matK* **	From/To	A	T	G	C
	A	-	*12.3839*	*5.8154*	**4.9319**
	T	*10.2083*	-	**5.8096**	*4.9368*
	G	*10.2083*	**12.3715**	-	*4.9368*
	C	**10.1981**	*12.3839*	*5.8154*	-
** *rbcL* **					
** *rbcL* **	From/To	A	T	G	C
	A	-	*9.6984*	*7.2905*	**7.2164**
	T	*9.1319*	-	**7.2832**	*7.2236*
	G	*9.1319*	**9.6887**	-	*7.2236*
	C	**9.1227**	*9.6984*	*7.2905*	-
** *ITS2* **					
** *ITS2* **	From/To	A	T	G	C
	A	-	*0.5673*	*1.7239*	**18.0123**
	T	*1.0720*	-	**45.5929**	*1.7201*
	G	*1.0720*	**15.0027**	-	*1.7201*
	C	**11.2258**	*0.5673*	*1.7239*	-

## Data Availability

Not applicable.

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
