# Peer review of "DNA Barcoding and ITS2 Secondary Structure Predictions in Taro (Colocasia esculenta L. Schott) from the North Eastern Hill Region of India"

_genes, 2022, doi:10.3390/genes13122294_

Round 1
Reviewer 1 Report
Mayengbam Premi et al. conducted an interesting study and I have comments below that will help improve the quality of the manuscript. I suggest authors to look through other published articles on plant DNA barcoding e.g., https://www.nature.com/articles/s41598-021-81087-w during the revision of this paper.
Title: The title seems ambiguous. Authors could reduce it to perhaps, DNA barcoding and ITS2 Secondary Structure of Taro (Colocasia esculenta L. Schott) from the North Eastern Hill Region of India.
Abstract
Line 26: Delete ‘have’.
Line 26-27: Authors should split this sentence into two comprehensive sentences
Line 30-32: This is method and not results. However, this is not a phylogenetic study but a sequence similarity tree estimation.
After Line: What is the implication of this study and further directions. I expect here that authors should elaborate on the use of multiple markers or even whole cp genome to study the plant for breeding purposes.
Introduction
Line 41: Delete ‘ which belongs to’ to ‘belonging to’
Line 43: Delete ‘ since time immemorial’ and ‘The history of taro is as old as its civilization.’
Line 45: Delete ‘time and again’ and just write ‘today’.
Line 46: Citation needed
Line 66: it should be: Phenotypic indicators have resolved…
Line 66-67. Authors should split this sentence into two comprehensive sentences
Line 69: Perhaps use ‘has been used to study taro genetic diversity’.
Line 76: This sentence could be confusing as one could misunderstand it. It could be more clearer if authors state that ‘Recent advancements in molecular and omics tools have help to overcome the limitations of morphological discriminations’
Line 81: ‘phylogenetic relationships’
Line 83: could validate the...
Line 86: can unveil…
Line 87: can help…
Line 92 – 114: I feel authors could summarize this paragraph because the introduction is already too long.
Line 119: Delete ‘aimed to assess’ and change it to ‘assessed’.
Methods
Line 129: I am wondering if this is meant to be a landrace instead of genotypes?
Line 153-154: What was the range of the initial total gDNA before it was adjusted to, and also where all DNA adjusted to exactly ng μL-1 or approximately? If it is approximate, authors should state it clearly here.
Line 174: delete ‘quantity of the’
Also, I guess authors checked the PCR products using gel electrophoresis? And this is usually done once after the PCR. However, it seems that authors performed the gel electrophoresis twice and I could not see the reason for performing the process more than once.
Results:
Section 3:1, what is the amplification success of each gene, approximate length of the genes, base constituents etc, are what I expect to see where.
Line 207-215, actually this type of data are not really included in modern day DNA barcoding paper and suggest deletion of this.
Line 218: Group of taxa? But this study dealt on one taxon!
Section 3.2 is just sequence similarity tree that was depicted and that is what the DNA barcoding entails and not evolutionary relationship. Please authors should carefully modify the result to reflect truly what DNA barcoding entails. I suggest authors to look through other published articles on plant DNA barcoding e.g., https://www.nature.com/articles/s41598-021-81087-w.
Lines 227 – 232 should be shifted to section 3.1
For the phylogram, this was unrooted and authors should clearly stated it in the caption for the figures. E,g., Unrooted Maximum Likehood tree …
The figure captions needs revision.
For instance, Figure 1 should be:
The unrooted maximum likelihood tree of the twenty taro genotypes based on matK gene sequence (Note: Bootstrap scores of ≥50% only are shown for each branch after 1000 bootstrap replication tests).
Use this pattern to correct other captions.
Authors should combine Table 1 and 2, leaving only the sample ID, Percent Identity, Accession number, species and gene of interest in the new table.
Section 3.3., what authors already did in section 3.2 is what DNA barcoding entails! Thus I do not see why it should be here. Again, I did not see Figure 4 in the text and cannot make comments based on that.
Discussion:
Line 282-283: I feel this sentence is untrue. DNA barcoding analysis is specifically for species identification. Remember that this involves the use of short fragment, and one genetic locus is not enough to infer evolutionary relationships. I feel lines 282 – 300 should be deleted entirely. Authors should focus mainly on discussing their results, comparing their findings with other studies as well as highlighting the implications of their studies and further directions. This should also extend to section 4.3 where most discussion here where most discussion here were merely superficial.
Author Response
Response to Reviewer 1 Comments
Reviewer’s comment: Mayengbam Premi et al. conducted an interesting study and I have comments below that will help improve the quality of the manuscript. I suggest authors to look through other published articles on plant DNA barcoding e.g., https://www.nature.com/articles/s41598-021-81087-w during the revision of this paper.
Authors’ response: Thank you very much for your kind remark and suggestion.
Point 1: Title: The title seems ambiguous. Authors could reduce it to perhaps, DNA barcoding and ITS2 Secondary Structure of Taro (Colocasia esculenta L. Schott) from the North Eastern Hill Region of India.
Authors’ response: Thank you for your suggestion. The title has been revised as ‘DNA barcoding and ITS2 Secondary Structure Predictions in Taro (Colocasia esculenta L. Schott) from the North Eastern Hill Region of India’.
Abstract
Point 2: Line 26: Delete ‘have’.
Authors’ response: Deleted ‘have’ in line 26 (revised line 28) as suggested.
Point 3: Line 26-27: Authors should split this sentence into two comprehensive sentences
Authors’ response: Thank you for your kind suggestion. The sentences have been modified and split into two sentences as suggested.
Point 4: Line 30-32: This is method and not results. However, this is not a phylogenetic study but a sequence similarity tree estimation.
Authors’ response: Thank you for your kind suggestion. Line 30 -32 have been deleted and modified according to the result obtained.
Point 5: After Line: What is the implication of this study and further directions. I expect here that authors should elaborate on the use of multiple markers or even whole cp genome to study the plant for breeding purposes.
Authors’ response: Thank you for your kind suggestion. We have updated the use of multiple markers and their implication in plant genetics and breeding as suggested.
Introduction
Point 6: Line 41: Delete ‘which belongs to’ to ‘belonging to’
Authors’ response: Revised as suggested.
Point 7: Line 43: Delete ‘since time immemorial’ and ‘The history of taro is as old as its civilization.’
Authors’ response: Thank you for your kind suggestion. The phrases ‘since time immemorial’ and ‘The history of taro is as old as its civilization.’ have been deleted as suggested.
Point 8: Line 45: Delete ‘time and again’ and just write ‘today’.
Authors’ response: Thank you for your kind suggestion. We have modified it as suggested.
Point 9: Line 46: Citation needed
Authors’ response: Thank you for your kind suggestion. We have cited the reference [2, Metthews and Ghanems, 2020] and updated in the reference section.
Point 10: Line 66: it should be: Phenotypic indicators have resolved…
Authors’ response: Thank you for your kind suggestion. We have revised it as suggested.
Point 11: Line 66-67. Authors should split this sentence into two comprehensive sentences
Authors’ response: The sentence has been split into two sentences as suggested.
Point 12: Line 69: Perhaps use ‘has been used to study taro genetic diversity’.
Authors’ response: The sentence has been modified, as suggested.
Point 13: Line 76: This sentence could be confusing as one could misunderstand it. It could be more clearer if authors state that ‘Recent advancements in molecular and omics tools have help to overcome the limitations of morphological discriminations’
Authors’ response: The sentence has been revised as suggested.
Point 14: Line 81: ‘phylogenetic relationships’
Authors’ response: Thank you for your observation. We have revised it as suggested.
Point 15: Line 83: could validate the...
Authors’ response: Thank you for your observation. We have revised it as suggested.
Point 16: Line 86: can unveil…
Authors’ response: Thank you for your kind suggestion. We have revised it as suggested.
Point 17: Line 87: can help…
Authors’ response: Thank you for your kind suggestion. We have revised it as suggested.
Point 18: Line 92 – 114: I feel authors could summarize this paragraph because the introduction is already too long.
Authors’ response: Thank you for your kind suggestion. We have revised it as suggested.
Point 19: Line 119: Delete ‘aimed to assess’ and change it to ‘assessed’.
Authors’ response: Thank you for your kind suggestion. We have revised it as suggested.
Methods
Point 20: Line 129: I am wondering if this is meant to be a landrace instead of genotypes?
Authors’ response: Thank you for your kind observation. Actually, the studied taro line includes some local landraces and advanced breeding lines (germplasms tracked for variety release), hence, we used the term ‘genotypes’ in a broad sense. The term ‘genotype’ may kindly be accepted.
Point 21: Line 153-154: What was the range of the initial total gDNA before it was adjusted to, and also where all DNA adjusted to exactly ng μL-1 or approximately? If it is approximate, authors should state it clearly here.
Authors’ response: The gDNA concentrations ranged from 300-500 ng μL-1 among the genotypes. It was adjusted exactly to 50 ng µL-1 by dilution following S1V1=S2V2.
Point 22: Line 174: delete ‘quantity of the’
Also, I guess authors checked the PCR products using gel electrophoresis? And this is usually done once after the PCR. However, it seems that authors performed the gel electrophoresis twice and I could not see the reason for performing the process more than once.
Authors’ response: Thank you so much for your kind observation. Actually, we have checked the quality of the PCR products and submitted to M/S Bioserve Biotechnologies (India) Pvt. Ltd, Hyderabad, India for the sequencing after a confirmation check. The sequencing provider has again performed a quality check before processing DNA sequencing.
We have revised the sentences to avoid confusion on duplicate electrophoresis.
Results:
Point 23: Section 3:1, what is the amplification success of each gene, approximate length of the genes, base constituents etc, are what I expect to see where.
Authors’ response: Thank you for your kind suggestion. As suggested, the details of the genes have been provided in Table 1.
Point 24: Line 207-215, actually this type of data are not really included in modern day DNA barcoding paper and suggest deletion of this.
Authors’ response: Thank you for your kind suggestion. We have deleted the paragraph as suggested. Subsequently, the supplementary figure S1 has been deleted.
Point 25: Line 218: Group of taxa? But this study dealt on one taxon!
Authors’ response: Thank you for your kind observation. We have deleted ‘the group of taxa’.
Point 26: Section 3.2 is just sequence similarity tree that was depicted and that is what the DNA barcoding entails and not evolutionary relationship. Please authors should carefully modify the result to reflect truly what DNA barcoding entails. I suggest authors to look through other published articles on plant DNA barcoding e.g., https://www.nature.com/articles/s41598-021-81087-w.
Authors’ response: Thank you very much for your kind remark and suggestion. We have followed the suggested article https://www.nature.com/articles/s41598-021-81087-w and other relevant papers and modified the results as suggested. The reference has also been cited in the reference section.
Point 27: Lines 227 – 232 should be shifted to section 3.1
Authors’ response: Thank you so much for your kind suggestion. The paragraph has been shifted from 3.2 to 3.1 as suggested.
Point 28: For the phylogram, this was unrooted and authors should clearly stated it in the caption for the figures. E,g., Unrooted Maximum Likehood tree …
The figure captions needs revision.
For instance, Figure 1 should be:
The unrooted maximum likelihood tree of the twenty taro genotypes based on matK gene sequence (Note: Bootstrap scores of ≥50% only are shown for each branch after 1000 bootstrap replication tests). Use this pattern to correct other captions.
Authors’ response: Thank you very much for your kind suggestion. The figure captions (Figures 1, 2, 3) have been revised as suggested.
Point 29: Authors should combine Table 1 and 2, leaving only the sample ID, Percent Identity, Accession number, species and gene of interest in the new table.
Authors’ response: Thank you very much for your kind suggestion. Table 1 and 2 have been combined into one table (Table 3) as suggested.
Point 30: Section 3.3., what authors already did in section 3.2 is what DNA barcoding entails! Thus I do not see why it should be here. Again, I did not see Figure 4 in the text and cannot make comments based on that.
Authors’ response: Thank you very much for your kind observation. We have revised the section 3.3 as ‘ITS2 Secondary Structure Predictions’. We extremely apologize for the misplacement of figure 4 due to the higher BMP file. We have inserted a compatible version of figure 4 (300dpi, tiff) in the text.
Discussion:
Point 31: Line 282-283: I feel this sentence is untrue. DNA barcoding analysis is specifically for species identification. Remember that this involves the use of short fragment, and one genetic locus is not enough to infer evolutionary relationships. I feel lines 282 – 300 should be deleted entirely. Authors should focus mainly on discussing their results, comparing their findings with other studies as well as highlighting the implications of their studies and further directions. This should also extend to section 4.3 where most discussion here where most discussion here were merely superficial.
Authors’ response: Thank you for your suggestion. The line 282-292 has been deleted as suggested. However, as rightly suggested in the previous comments to highlight the benefit of multiple markers, we retained line 292-300. As reports on DNA barcoding in Colocasia are scanty, the discussion has been updated with recent studies on yam (Dioscorea spp.) and Vicia species in lines 345 – 361.
Thank you very much for your constructive suggestions which enabled us to revise the manuscript to this extent.
Reviewer 2 Report
The paper by Premi Devi et al. entitled Genotypic discrimination in Taro (Colocasia esculenta L. Schott) Indigenous to North Eastern Hill Region of India: Insights into DNA Barcoding, ITS2 Secondary Structure Predictions, and Molecular Phylogeny deals with an important topic which is the genotypic variability of an underutilized crop which is Taro roots.
Generally the paper is well written however I have noticed some mistakes highlighted in the attached file and here is a detailed assessment of the paper
Abstract: can be improved by inserting a sentence related to the limit of the present study
Introduction section: For this section I suggests that authors reduce the length of the part (L92-L115) by concentrating on the limit of the previous studies and the poposed novelty of the present study over them clearly stated.
Materials and Methods : This section is accurately described
Results: only some mistakes highlighted that should be corrected and considered
Discussion: you can delete what was already inserted previously and concentrate on the discussion of the results here
The results should be also compared with previous work on the same or other crops using the same technique analytical tool
Conclusions: the section is poor here and needs to include the limit of the present study and where future studies on the same crop and/or with similar abjective should preferably focus based on this study
References : some mistakes to be corrected

Author Response
Response to Reviewer 2 Comments
Point 1: The paper by Premi Devi et al. entitled Genotypic discrimination in Taro (Colocasia esculenta L. Schott) Indigenous to North Eastern Hill Region of India: Insights into DNA Barcoding, ITS2 Secondary Structure Predictions, and Molecular Phylogeny deals with an important topic which is the genotypic variability of an underutilized crop which is Taro roots.
Generally the paper is well written however I have noticed some mistakes highlighted in the attached file and here is a detailed assessment of the paper
Authors’ response: Thank you very much for your kind remarks.
Point 2: Abstract: can be improved by inserting a sentence related to the limit of the present study
Authors’ response: Thank you very much for your kind suggestion.
‘Taro is highly polygenic, polyphyletic, and poly geographic in nature, which leads to its rapid genetic erosion due to spontaneous mutation of vegetative planting materials. To prevent the perceived loss of taro diversity, species discrimination and genetic conservation of promising taro genotypes need special attention’, which we have mentioned in the abstract. In addition, we have incorporated ‘Reports on genetic discrimination of taro at its centre of origin is still untapped’ as suggested by the reviewer.
Point 3: Introduction section: For this section I suggests that authors reduce the length of the part (L92-L115) by concentrating on the limit of the previous studies and the poposed novelty of the present study over them clearly stated.
Authors’ response: Thank you very much for your kind suggestion. We have shortened the paragraph by deleting lines 114-118 as suggested by the reviewer 1. The novelty and aim of the study have been updated line 124-136.
Point 4: Materials and Methods: This section is accurately described
Authors’ response: Thank you very much for your kind remarks.
Point 5: Results: only some mistakes highlighted that should be corrected and considered
Authors’ response: The highlighted portions have been modified accordingly.
Point 6: Discussion: you can delete what was already inserted previously and concentrate on the discussion of the results here
Authors’ response: Thank you very much for your suggestion. The previous described portions have been deleted from the discussion part, as suggested by both esteemed reviewers (Lines 282-292).
Point 7: The results should be also compared with previous work on the same or other crops using the same technique analytical tool
Authors’ response: Thank you very much for your kind suggestion. As reports on DNA barcoding in Colocasia are scanty, the discussion has been updated with recent studies on yam (Dioscorea spp.) and Vicia species in lines 345 – 361.
Point 8: Conclusions: the section is poor here and needs to include the limit of the present study and where future studies on the same crop and/or with similar abjective should preferably focus based on this study
Authors’ response: The conclusion section has been updated with future studies and focused on similar investigation, as proposed by the reviewer.
Point 9: References: some mistakes to be corrected
Authors’ response: We have cross-checked the references and revised as suggested.
Thank you very much for your constructive suggestions which enabled us to revise the manuscript to this extent.
Round 2
Reviewer 2 Report
The authors have done a good editing to the whole manuscript
I recommend to accept it as it is